**PLOS** NEGLECTED TROPICAL DISEASES

# Opisthorchis viverrini, Clonorchis sinensis and Opisthorchis felineus liver flukes affect mammalian host microbiome in a species-specific manner

**Maria Y. Pakharukova**[1,2]*, **Ekaterina A. Lishai**[1,2], **Oxana Zaparina**[1], **Nina V. Baginskaya**[1], **Sung-Jong Hong**[3], **Banchob Sripa**[4], **Viatcheslav A. Mordvinov**[1]

1 Institute of Cytology and Genetics, Siberian Branch of Russian Academy of Sciences, Novosibirsk, Russia, 2 Department of Natural Sciences, Novosibirsk State University, Novosibirsk, Russia, 3 Convergence Research Center for Insect Vectors, Incheon National University, Incheon, Korea, 4 WHO Collaborating Centre for Research and Control of Opisthorchiasis (Southeast Asian Liver Fluke Disease), Tropical Disease Research Center, Department of Pathology, Faculty of Medicine, Khon Kaen University, Khon Kaen, Thailand

* pmaria@yandex.ru

**Data Availability Statement:** All data generated or analyzed during this study are included in this published article and its Supporting Information

## Abstract

### Background

Opisthorchis felineus, Opisthorchis viverrini and Clonorchis sinensis are epidemiologically significant food-borne trematodes endemic to diverse climatic areas. O. viverrini and C. sinensis are both recognized to be 1A group of biological carcinogens to human, whereas O. felineus is not. The mechanisms of carcinogenesis by the liver flukes are studied fragmentarily, the role of host and parasite microbiome is an unexplored aspect.

### Methodology/Principal findings

Specific pathogen free Mesocricetus auratus hamsters were infected with C. sinensis, O. viverrini and O. felineus. The microbiota of the adult worms, colon feces and bile from the hamsters was investigated using Illumina-based sequencing targeting the prokaryotic 16S rRNA gene. The analysis of 43 libraries revealed 18,830,015 sequences, the bacterial super-kingdom, 16 different phyla, 39 classes, 63 orders, 107 families, 187 genera-level phylotypes. O. viverrini, a fluke with the most pronounced carcinogenic potential, has the strongest impact on the host bile microbiome, changing the abundance of 92 features, including Bifidobacteriaceae, Erysipelotrichaceae, [Paraprevotellaceae], Acetobacteraceae, Coriobacteraceae and Corynebacteriaceae bacterial species. All three infections significantly increased Enterobacteriaceae abundance in host bile, reduced the level of commensal bacteria in the gut microbiome (Parabacteroides, Roseburia, and AF12).

### Conclusions/Significance

O. felineus, O. viverrini, and C. sinensis infections cause both general and species-specific qualitative and quantitative changes in the composition of microbiota of bile and colon feces of experimental animals infected with these trematodes. The alterations primarily concern

files. Raw sequencing data have submitted to the NCBI database (BioProject PRJNA866652) and will be freely accessible if the manuscript is accepted for publication.

**Funding:** Library preparation, sequencing, and data analysis was financially supported by the Russian Science Foundation and Novosibirsk Oblast (https://rscf.ru/en/) (grant number 22-24-20010 to M.Y.P.). The microscopic analysis was conducted at the Microscopy Center of the ICG SB RAS and was supported by the Siberian Branch of the Russian Academy of Science (grant number FWNR-2022-0021 to V.A.M.). The funders had no role in study design, data collection and analysis, decision to publish, or preparation of the manuscript.

**Competing interests:** The authors declare no competing interests.

the abundance of individual features and the phylogenetic diversity of microbiomes of infected hamsters.

## Author summary

Three epidemiologically significant food-borne trematodes (*Opisthorchis felineus*, *O. viverrini*, *Clonorchis sinensis*) affect the hepatobiliary system of mammals, including humans, inducing cholangitis, bile duct neoplasia, and even cholangiocarcinoma among chronically infected individuals. Two species, *O. viverrini* and *C. sinensis* are both recognized 1A group of biological carcinogens to human, whereas *O. felineus* is classified as a noncarcinogen. The impact of microbiota in the differences in morbidity for these three infections is not clear due to the absence of comparative studies conducted within the same experimental setting.

Here we examined the microbes of liver flukes *C. sinensis* (South Korea), *O. viverrini* (Thailand) and *O. felineus* (Russia) as well as that in the bile and feces of hamsters that were infected with these three species of worms.

The liver flukes (*O. felineus*, *O. viverrini*, and *C. sinensis*) contain their own unique bacteria and contribute to significant species-specific alterations in gastrointestinal microbiota of the mammalian host. *O. viverrini*, a fluke with the most pronounced carcinogenic potential, has the strongest impact on the bile microbiome. *O. felineus* has the smallest consequences both in bile and feces microbiome. Our data can lead to the development of more effective species-specific modalities for diagnosis, prevention, and treatment.

## 1. Introduction

Helminthic infections remain a global public health issue, particularly in low- and middle-income countries [1]. Helminthic infections cause significant changes in microbiota composition [2–4]. These changes may contribute to helminth-induced morbidity and carcinogenesis [5–7]. Food-borne Opisthorchiidae trematodes *Opisthorchis viverrini* (Poirier, 1886), *Clonorchis sinensis* (Cobbold, 1875), and *Opisthorchis felineus* (Rivolta, 1884) infest the hepatobiliary system of mammals, including humans [8–9], affect ~40 million people worldwide, have a major socioeconomic impact in endemic areas [10], and have dramatic carcinogenic consequences.

*O. felineus* is endemic to North Asia and Eastern Europe, opisthorchiasis viverrini is largely confined to Southeast Asia [10], and clonorchiasis is found in a number of countries in the Far East: Korea, parts of Russia, China, and Vietnam [11–12]. These hepatic helminthiases share similar clinical features and are treated with the same anthelmintics, thereby giving an impression that there is no difference among these infections. At the same time, there are data indicating that each of these trematodiases also has its own distinct clinical manifestations.

The liver flukes have a complex life cycle with alternation of two intermediate hosts and one definitive host. Infection of mammals occurs as a result of eating raw or undercooked fish [8, 10–13]. After entering the digestive tract of the final host, excysted metacercariae reach the biliary tract. Upon reaching maturity, the parasites produce a large amount of eggs containing miracidia: a form invasive for the first intermediate host, freshwater gastropods. In the snail, trematodes go through asexual stages of development and form cercariae, a free-living stage.

The cercariae leave the snail and infect the second intermediate host, mainly cyprinid fish. In fish, cercariae encapsulate and develop into metacercariae, which are the only stage infectious for mammals.

Notably, different species of liver flukes have adapted to specific features of endemic foci, including the range of primary, secondary, and definitive hosts [12–14]. For instance, reservoir hosts for *O. felineus* are mainly wild mammals, and the role of humans is not essential. In contrast, *O. viverrini* foci are considered to be of anthropogenic origin, while *C. sinensis* foci of mixed origin are regarded as zoonotic-anthropogenic [14]. In addition, the first stages of the life cycle of various species of opisthorchids take place in water bodies with specific microflora at substantially different temperatures and other climatic conditions.

The most striking example of differences among the liver flukes is their carcinogenicity to humans. Asian liver flukes—*O. viverrini* and *C. sinensis*—are considered the main cause of bile duct cancer (cholangiocarcinoma; CCA) in endemic areas and have been classified as Group 1 biological carcinogens, whereas *O. felineus* is classified as a noncarcinogen (3A Group) [8]. Long-term observational data indicate that CCA is recorded significantly less frequently on the territory of *O. felineus* foci than among residents of endemic areas of *O. viverrini* and *C. sinensis* [9, 12–13].

Helminthiases associated with malignancy remain largely unexplored [15]. Could differences in the microbiome among the three opisthorchid species be the reason for the different carcinogenicity of these closely related species? Is there any reason to believe that the opisthorchid microbiota contributes to the differences in morbidity? What changes occur in the bile and gut microbiome of mammals infected with liver flukes? Unfortunately, the existing data are insufficient to answer these questions.

One of possible explanations for this state of affairs is the lack of comparative analyses—of microbiomes of opisthorchids and model animals infected with different fluke species—conducted within the same experimental setting in order to minimize potential between-study discrepancies.

In this work, we aimed to evaluate possible differences in the microbiome among *O. viverrini*, *O. felineus*, and *C. sinensis* and hamsters infected with these flukes in one laboratory setting. For this purpose, we infected specific pathogen free (SPF) hamsters with *C. sinensis* metacercaria from South Korea, with *O. viverrini* metacercaria from Thailand, or with *O. felineus* metacercaria from Russia. One month after the infection, we characterized the microbial communities after sequencing of libraries constructed from the V3-V4 region of 16S ribosomal DNA from adult worms and from colon feces and bile of the hamsters.

As our knowledge about fundamental characteristics and biology of the helminth infection–associated host microbiome grows, so will (i) our understanding of species-specific mechanisms of host–parasite interaction in opisthorchiasis and clonorchiasis and (ii) the quality of treatment of these diseases, while their complications will be ameliorated.

## 2. Methods

### 2.1. Ethics statement

All the procedures were in compliance with EU Directive 2010/63/EU for animal experiments. Study design protocols and standard operating procedures (concerning the hamsters and the fish) were approved by the Committee on the Ethics of Animal Experiments at the Institute of Cytology and Genetics, Siberian Branch of Russian Academy of Sciences (ICG SB RAS) (permit number 126 of 3 August 2022).

## 2.2. Experimental design

SPF golden Syrian hamsters *Mesocricetus auratus* from the SPF animal facility at the Institute of Cytology and Genetics, Siberian Branch of Russian Academy of Sciences (ICG SB RAS) were used in this study as generally accepted laboratory model animals [7, 10–13]. All the procedures were performed aseptically. We applied an appropriate randomization strategy (blocking) to control possible variables, such as potential infection, among the experimental animals. We took into account nuisance variables that could bias the results (a litter and an investigator).

For collecting *O. felineus* metacercariae, a naturally infected freshwater fish (*Leuciscus idus*) was net-caught in the Ob River near Novosibirsk (Western Siberia, Russia) by research assistant Viktor Antonov (ICG SB RAS) without the use of chemicals. *O. felineus* metacercariae were extracted as described previously [7, 16]. *C. sinensis* and *O. viverrini* metacercariae were extracted from naturally infected freshwater fish (Seoul, Republic of Korea, and Khon Kaen, Thailand, respectively) and delivered on ice. After several washes with normal saline, metacercariae were identified under a light microscope. All the procedures with hamsters were performed at the SPF animal facility at the ICG SB RAS.

Sixteen hamsters were distributed into four groups, and animals from three of them were infected with 75 metacercariae (of one of the three liver fluke species separately) by gastric intubation at intervals of 3–5 days to avoid bacterial cross-infection. One group was kept uninfected. One month after the infection, the hamsters were euthanized using carbon dioxide. All the procedures were done aseptically. Bile samples were collected via puncture of the gall bladder and stored at -80°C until use. Colorectal feces were extracted and stored at -80°C until use. Adult worms were carefully extracted from the biliary tract, washed more than 10 times with sterile saline, then soaked for several hours in sterile saline at 37°C, and finally stored at -80°C until analysis.

Although all procedures with hamsters were carried out in the same Animal Facility, we cannot exclude any small differences in the standard protocol for the metacercariae isolation from fish that might affect the microbiome.

## 2.3. DNA extraction

Feces (150–200 mg) and bile (50–100 μl) samples were subjected to DNA extraction [7]. The feces were washed with 1 ml of 20% ethanol, homogenized, and then washed twice with 20% ethanol and three times with 1 ml of PBS. The samples were boiled at 100°C (for 1 min) followed by 4 freeze–thaw cycles using liquid nitrogen and boiling water (1 min each procedure). The samples were treated with 0.2 mg/ml proteinase K (Thermo Scientific, USA) for 1 h at 56°C, and then DNA was extracted by the phenol–chloroform method. Additionally, DNA samples were purified with the DU10 Kit (Biolabmix, Russia).

DNA from whole adult worms was extracted by the standard method based on proteinase K digestion and phenol–chloroform extraction. Frozen worms (10 to 15 mg) were subjected to DNA extraction. DNA concentrations were measured on a Qubit 2.0 spectrophotometer (Invitrogen, USA). The purity of DNA was assessed by measuring absorbance at 260 and 280 nm on a NanoDrop 2000 spectrophotometer (NanoDrop Technologies, Wilmington, DE, USA); ratios of these values ranging from 1.7 to 2.1 were considered acceptable.

## 2.4. Library preparation and sequencing

Amplicons containing the V3-V4 hypervariable region of the 16S rRNA gene were obtained by PCR using standard gene-specific primers (bases are boldfaced) for the 16S rRNA gene with TrueSeq adapters:

S-D-Bact_0341_F: ACACT CTTTC CCTAC ACGAC GCTCT TCCGA **TCTCC TACGG GNGGC WGCAG** and

S-D-Bact_0785_R: GACTG GAGTT CAGAC GTGTG CTCTT CCGAT CTG**AC TACHV GGGTA TCTAA TCC**.

The thermal cycling conditions were as follows: We aimed to minimize PCR cycles, which meant using different numbers of cycles for worms (10), feces (20), and bile (25–30 cycles).

The first step was melting at 95°C for 2 min, followed by the second step (95°C 15 s, 55°C 25 s, 72°C 30 s) for 10 cycles for DNA from worms, bile, and feces; the third step was 15 cycles of 95°C 15 s, 60°C 20 s, and 72°C 30 s for DNA from bile or 10 cycles for DNA from feces.

Negative control samples were set up by adding equal volume of water (Water Molecular Biology Grade, PanReac AppliChem) instead of genomic DNA and, although there were no visible bands on a gel, they were processed the same way as DNA from bile.

To isolate a desired DNA fragment and get rid of nonspecific products, amplicons were purified by size selection in a 0.7% SeaKem GTG Agarose gel (cat. #: 50071; Lonza, Switzerland). DNA concentration was measured by means of Qubit 2.0 (Qubit dsDNA HS Assay Kit, Invitrogen, Thermo Fisher Scientific, USA).

The samples were next subjected to a second PCR round where PCR products were barcoded. NEBNext Multiplex Oligos for Illumina (Dual Index Primers Set 2, cat. #: E7780S, New England Biolabs, UK) were employed for the barcoding.

The thermal cycling program consisted of one cycle of 95°C for 3 min; followed by eight cycles of 95°C for 30 s, 55°C for 30 s, and 72°C for 30 s; with a final extension cycle of 72°C for 5 min (six cycles of PCR) (S1 Fig). Size selection of DNA was performed on Agencourt AMPure XP beads (Beckman Coulter). The size and quantity of the libraries were verified on an Agilent Bioanalyzer (S2 Fig).

The obtained libraries were analyzed by paired-end (2 × 300 base pairs) sequencing on the MiSeq platform (Illumina) at the BGI Genomics (China, Hong Kong) using the MiSeq v3 Reagent Kit (Illumina). Forty-three DNA libraries were obtained, including two negative control samples (S3 Fig). Output statistics of raw sequencing data are presented in S4 Fig.

## 2.5. Bioinformatics analysis

Examination of the 16S rRNA gene reads was conducted by means of the QIIME2 (quantitative insight into molecular ecology) pipeline https://qiime2.org/ (Table A in S1 Tables) [17]. Feature picking and filtering of chimeric sequences were carried out in QIIME2 using the DADA2 algorithm. Chimera-checked Greengenes taxonomy served as a reference for taxonomic assignment (Greengenes v.13.8, https://greengenes.secondgenome.com/). After taxonomic assignment and demultiplexing, alpha diversity within and between groups was calculated in QIIME2 using Observed features, Shannon's, Fisher's, Brillouin's, and Faith's alpha metrics at a depth of 20000 sequences per sample. Alpha diversity comparisons were performed by the Kruskal–Wallis nonparametric test and 999 permutations ('stats' version 3.6.2 R package). Beta diversity was investigated by principal coordinate analysis (PCoA) both on non-normalized and normalized (CSS algorithm [18]) data with the help of unweighted and weighted UniFrac distance (implemented in QIIME2). A phylogenetic tree was constructed in QIIME2 (align-to-tree-mafft-fasttree) using the q2-phylogeny plugin and was visualized in MEGA-X v.10.2.5 https://www.megasoftware.net/citations [19].

To examine significance of variation among groups at the feature level, we utilized the zero-inflated Gaussian distribution model (fitZig model) and the presence-absence test ('metagenomeSeq' R-package https://www.bioconductor.org/packages/release/bioc/html/metagenomeSeq.html [18]). To assess significance of variation among groups at phylum, class,

order, family, and genus levels, we applied the aggregateByTaxonomy function (aggTax) ('meta-genomeSeq', R-package) and the nonparametric Wilcoxon signed-rank test or Kruskal Wallis test followed by post-hoc Dunn's test to CSS-normalized data ('stats' version 3.6.2; 'FSA" version 0.9.3 R-packages) [20]. To adjust data for the false discovery rate during multiple comparisons, the Benjamini–Hochberg correction or Bonferroni correction was employed ('stats' v. 3.6.2 R-package). The pipeline of implemented steps is presented in Table A in S1 Tables.

## 2.6. Immunohistochemistry

Immunohistochemical analysis was performed as described elsewhere [21–23]. For this assay, livers were immediately fixed in 4% paraformaldehyde in phosphate-buffered saline (PBS) for 12 h, then soaked in 30% sucrose at 4˚C for 24 h, and stored at -80˚C until use. The livers were sliced at 7 or 10 μm thickness on a Microm HM-505 N cryostat. Staining was conducted by means of goat polyclonal antibodies against Lipid A of Enterobacteriaceae bacteria (cat. #: PA1-28903, Invitrogen, USA, 1:200), followed by probing with fluorescently labeled secondary antibodies (Abcam, USA). The slices were treated with the Phalloidin-iFluor 488 Reagent (cat. #: ab176753, Abcam), coverslipped with the Fluoro-shield mounting medium containing 4′,6-diamidino-2-phenylindole (DAPI; cat. # F6057, Sigma-Aldrich, USA), and visualized under an AxioImager A1 microscope (Zeiss) with camera AxioCam MRc (Zeiss).

## 3. Results

### 3.1. All samples

The sequencing analysis of 43 libraries (S3 Fig) revealed 18,830,015 sequence reads: 5,467,250 in samples from worms (29%), 8,129,188 in samples from feces (43%), 4,375,296 in samples from bile (23%), and 858,281 in negative control samples (5%). On average, there were 427,955 reads per sample (median: 410,272). The lowest number of sequences was found in a negative control sample (14,249), and the largest in the sample from the *C. sinensis* worm (909,445).

An average of 1% unclassified sequences (maximum 4%) were found (listed in the Green-genes database as "Unassigned, k__Bacteria" and "k__Bacteria; p__"). Among these sequences, using BLASTn, sequences were found corresponding to the species of *Sphingomonas*, Bacteroidetes phylum, *O. viverrini*, and the mitochondrial 12S rRNA of *M. auratus*.

Of the 18,830,015 sequence reads, 11,644,623 remained after DADA2 processing (Table B in S1 Tables) for taxonomic and diversity analysis. They were assigned to 18,725 features (amplicon sequence variants), including 102 features in the negative control samples. Finally, 17,644 features were classified when we removed all unassigned ones ("Unassigned, k__Bacteria" and "k__Bacteria; p__"). The average fragment length was 414 nucleotides, with a minimum of 264 and a maximum of 478 nucleotides.

Taxonomic composition was as follows: the bacterial super-kingdom, 16 different phyla, 39 classes, 63 orders, 107 families, and 187 genera level phylotypes that were well supported (Table C in S1 Tables).

### 3.2. Diversity of all microbiomes

To look at the overall picture of the distribution of microbiological communities among the samples, we analyzed the diversity of all samples by categorizing them by the types of sources from which DNA was isolated: feces, bile, and parasites. Analysis of alpha diversity, that is, of taxonomic richness of communities, based on the number of observed features (amplicon sequence variant) (Fig 1A) and a phylogenetic diversity (Faith's PD) index (Fig 1B) revealed greater alpha diversity in the worm samples than in the feces (P = 5E-5) and bile (P = 0.0001)

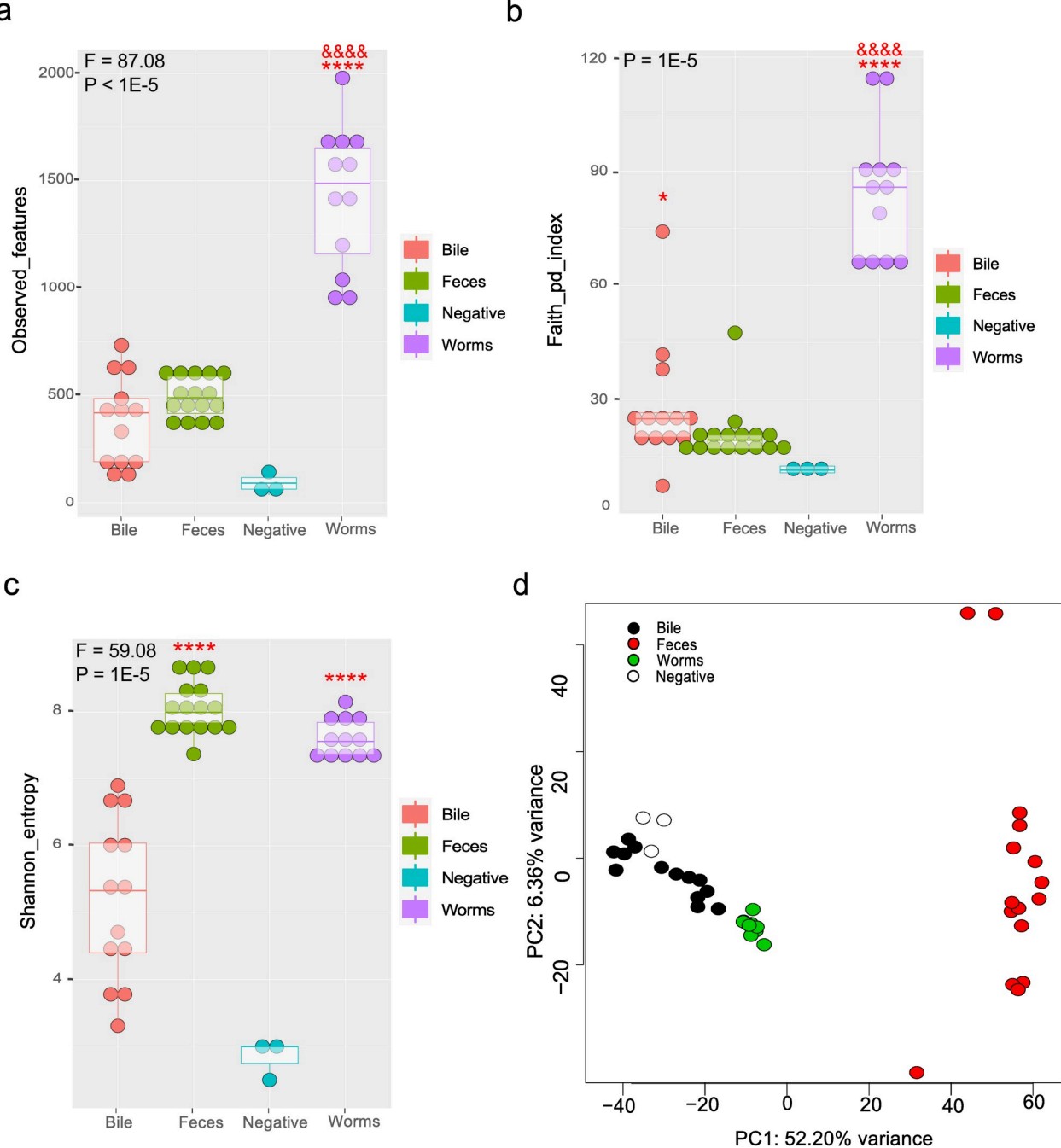

**Fig 1. Alpha and beta diversity levels of the DNA libraries prepared from feces, bile, and worms.** a. Observed features. b. Phylogenetic diversity index (Faith's PD). c. Shannon's index. d. Principal coordinates (Bray–Curtis distances) of DNA libraries. $^{*}P_{adj} < 0.05$ as compared to fecal samples, $^{****}P_{adj} < 0.001$ as compared to bile samples, $^{\&\&\&\&}P_{adj} < 0.001$ as compared to bile samples.

samples. Taxonomic richness in bile communities was also higher than that in fecal samples (Faith's PD, P = 0.02).

Another alpha diversity index, which takes into account not only taxonomy but also the evenness of each feature (Shannon's index; Fig 1C), revealed higher diversity in both feces (P < 0.0001) and worms (P < 0.0001) in comparison with bile samples. These results are

consistent with previously published data, where a high diversity index was found in fecal microbiomes [4].

PCoA using a Bray–Curtis dissimilarity matrix (Fig 1D) showed a significant clustering of samples depending on the source: bile and feces of hamsters as well as worms. Pairwise permutation analysis of variance by means of the same matrix revealed that all groups significantly differ from each other (q-value for each pair of samples was 0.001). This allowed further analysis of the groups separately.

In fecal samples, nine major phyla of bacteria were found, which were also present in samples from bile and worms (e.g., Bacteroidetes and Firmicutes). Bile samples contained 15 phyla of bacteria, and 15 phyla were found in worm samples (Fig 2A and 2B).

Major bacterial phyla (e.g., Proteobacteria, Firmicutes, Actinobacteria, and Bacteroidetes) in bile feces samples were the same among all samples and matched the microbiome of feces (Fig 2A–2C). The infected bile also contained the worm microbiota (minor phyla); these taxa were absent in uninfected animals and are listed in Fig 2B.

The structure of the fecal microbiome is different from that of worms and bile (Fig 2C). The predominant phyla of bacteria in the fecal microbiome are Bacteroidetes and Firmicutes (~50% relative abundance for each type), while in the microbiome of bile and worms, the Proteobacteria phylum is the most represented (~50%), and Bacteroidetes and Firmicutes make up ~40% together. Besides, ~10% of the microbial communities of bile and worms are bacteria of the Actinobacteria phylum.

The major phyla were represented by four phyla in all three helminths: Proteobacteria, Bacteroidetes, Firmicutes, and Actinobacteria, with relative abundance levels of 49%, 23%, 16%, and 8%, respectively. All other phyla of bacteria are present at less than 1–2%. Meanwhile, bile microbial communities were found to be similar to those of helminths, in particular, the matching phyla are the four major phyla, with relative abundance levels of 60%, 29%, 7%, and 3%, respectively (Fig 2A–2C).

Fig 3 demonstrates a phylogenetic tree constructed at the level of a class. Those classes of bacteria that were found in the infected animals or in worm samples and are not typical for communities of uninfected animals are highlighted in red.

### 3.3. Bile microbiomes

A total of 2004 features were identified in the bile samples. Taxonomic composition was as follows: the bacterial superkingdom, 15 different phyla, 30 classes, 93 families, and 187 genera.

The top five most abundant families in uninfected bile were Sphingomonadaceae, S24-7, Chitinophagaceae, Nocardiaceae, and Caulobacteraceae. In comparison, there were some changes in the bile of infected animals: Erysipelotrichaceae relative abundance increased significantly after the *C. sinensis* or *O. viverrini* infection. In the *O. viverrini*–infected bile, Erysipelotrichaceae was among the five major families of bacteria.

Meanwhile, the top five most abundant genera in bile samples were *Sphingomonas*, *Sediminibacterium*, *Ruminococcus*, *Allobaculum*, and *Rhodococcus*, with relative abundance levels of 18.4–20.1%, 9.0–9.8%, 1.9–2.6%, 3.7–4.0%, and 1.9–2.6%, respectively (S5 Fig).

Analysis of the index of phylogenetic diversity (Faith's PD; P value = 0.12) showed an insignificant increase within the bile microbiome after the infection (Fig 4A).

Shannon's index, taking into account the number of species and the evenness of the species distribution, was the highest in bile samples from *O. viverrini*–infected hamsters and was significantly different from that in bile from the uninfected animals (Fig 4B). When evaluating beta diversity by pairwise permutation analysis of variance using the Bray–Curtis dissimilarity matrix, we found no significant differences (Fig 4C).

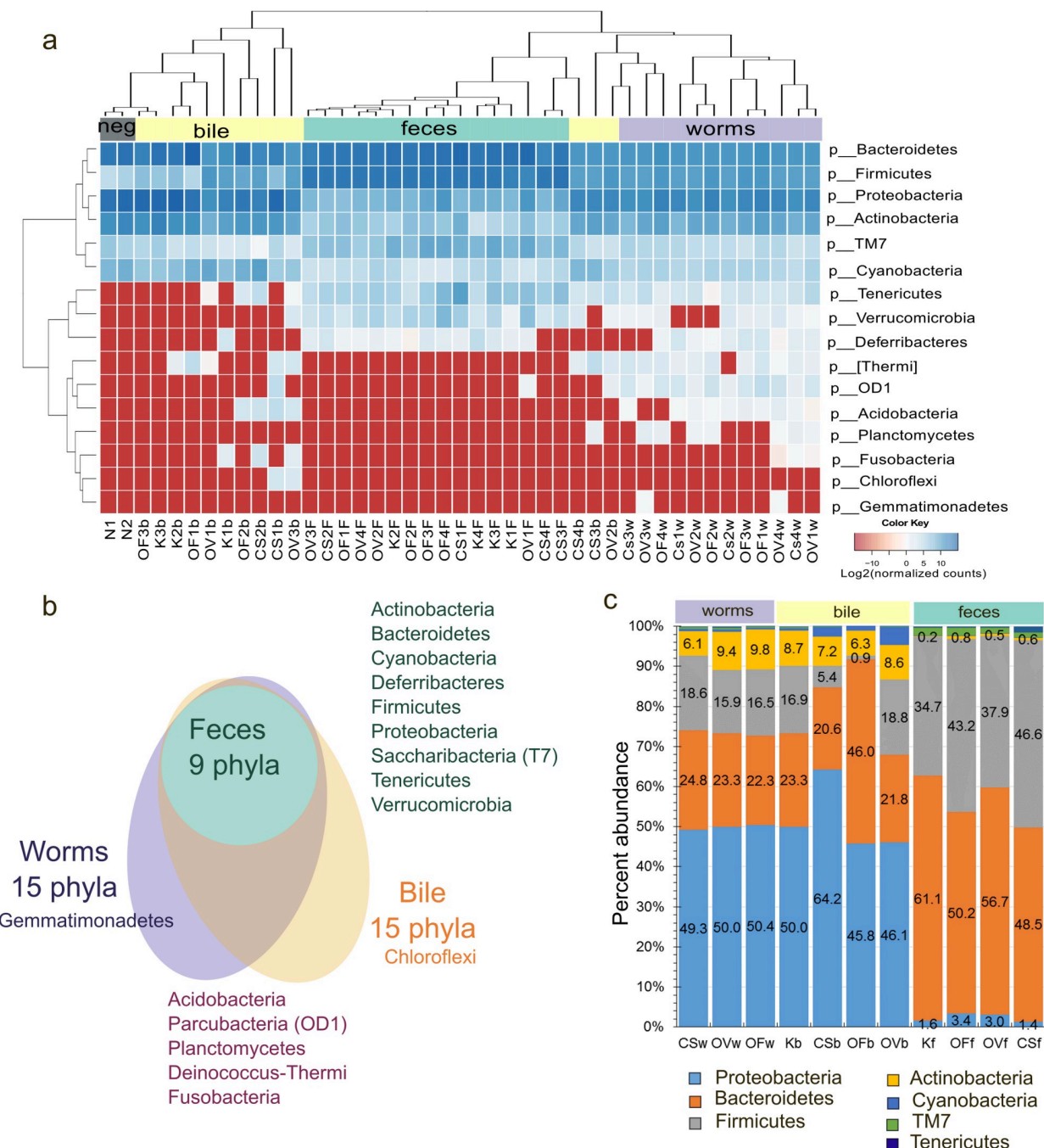

**Fig 2. Taxonomic diversity of the samples prepared from feces, bile and worms.** a. The heatmap illustrating relative abundance of taxa at the phylum level. Neg: negative control samples (no-DNA control). b. The Venn diagram of phyla distribution among samples of feces, bile of hamsters, and worms. c. The boxplot of the distribution of seven major bacterial phyla from microbial communities of worms, feces, and bile of hamsters. The average percent abundance is presented. K: uninfected animals; N: negative control; OF: *O. felineus*, CS: *C. sinensis*, OV: *O. viverrini*, f: feces, b: bile, w: worms.

It is noteworthy that when comparing samples of infected bile and uninfected bile at the genus level, we identified 68 infection-associated genera (Fig 4D). Further analysis of these genera indicated that 26 of them were also found in worm samples, and 42 of these infection-associated genera were absent in helminths. It is possible that the 42 genera are secondary

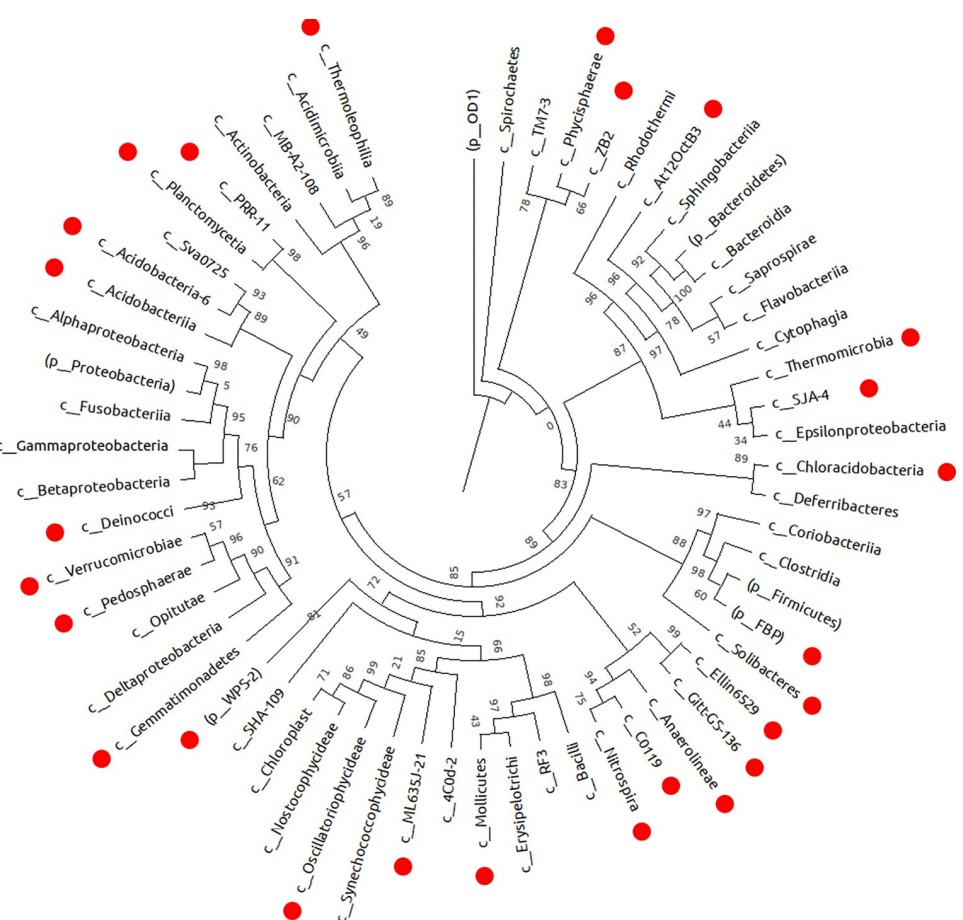

**Fig 3. The phylogenetic tree of microbial communities at the class level.** Red circles denote classes of bacteria that were found in the infected animals or in worm samples but not in the samples from uninfected animals.

infections, e.g., *Burkholderia*, *Phenylobacterium*, *Blastomonas*, and *Serratia*. Meanwhile, 26 genera emerged from microbiome exchange with helminths; these are such genera as *Azospira*, *Klebsiella*, *Haemophilus*, *Finegoldia*, and *Deinococcus* (Figs 4D, 5A and 5B).

ZigFit analysis revealed 12 features differentially present ($P_{adj} < 0.05$) in the infected bile samples compared to uninfected bile samples. Relative abundance of four of them significantly diminished after the infection ($\log_2$ fold change $< -6$), and for eight features, relative abundance significantly increased ($\log_2$ fold change $> 6$) ($P_{adj} < 0.05$; Table D in S1 Tables). Among the downregulated features, there were *Sediminibacterium*, *Chryseobacterium*, and *Janthinobacterium*. Among features whose abundance increased, there were for example *Acinetobacter johnsonii*, Enterobacteriaceae, and *Achromobacter spp*. The same result was obtained in the presence-absence test: 10 features significantly differentially present in the infected bile samples compared to uninfected bile samples (e.g., *Acinetobacter johnsonii*, *Achromobacter*, *Sediminibacterium*, *Ralstonia*, and *Janthinobacterium* genera) and Enterobacteriaceae family ($P_{adj} < 0.05$; Table E in S1 Tables). Regardless of helminth species, we found a significant increase in relative abundance of Erythrobacteraceae and Enterobacteriaceae.

Meanwhile, significant differences were observed in the bile microbiome depending on species; for example, *O. felineus* infection significantly changed 36 features ($P_{adj} < 0.05$, with 14 features downregulated and 22 upregulated) (Table E in S1 Tables), *C. sinensis* infection significantly changed 55 features ($P_{adj} < 0.05$, with 5 features downregulated and 50 upregulated)

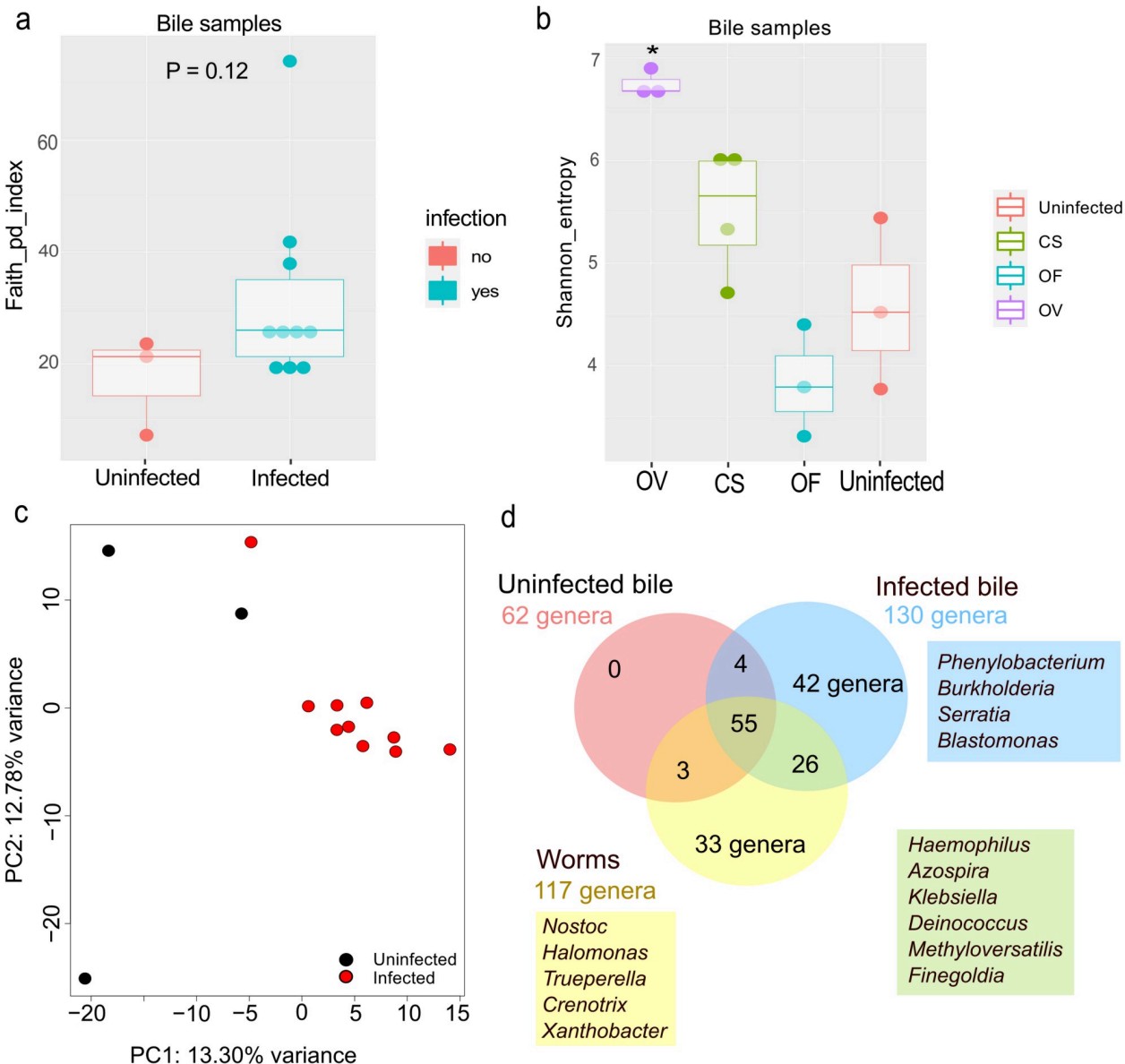

**Fig 4. Results of alpha and beta diversity analyses of the bile samples.** a. Differences in the index of phylogenetic diversity (Faith's PD) between uninfected and infected samples of bile. b. Shannon's index. OF: *O. felineus*, CS: *C. sinensis*, OV: *O. viverrini*. $^*P_{adj} < 0.05$, significant changes as compared to the uninfected samples (Kruskal–Wallis test). c. The results of PCoA (with a metric of Bray–Curtis distances) of DNA libraries. d. The Venn diagram constructed at the level of genera for communities of three types of samples: infected bile (blue circle), uninfected bile (red circle), and parasites (yellow circle).

(Table G in S1 Tables). *O. viverrini* infection changed the largest number of taxa: 92 features ($P_{adj} < 0.05$, with 20 features downregulated and 72 upregulated) (Table H in S1 Tables).

There were 14 differentially distributed taxa in the Kruscall-Wallis test on the genus level and eleven taxa on the family level among bile samples (Fig 5A–5D) ($P_{adj} < 0.05$; Table I in S1 Tables). They were Acetobacteraceae, Bifidobacteraceae, Coriobacteraceae, Corynebacteriaceae, Enterobacteriaceae, Microbacteriaceae (Fig 5A), Erysipelotrichaceae (Fig 5B), Paraprevotellaceae, S24-7, Sinobacteraceae, and Ruminococcaceae. At the genus level, significant differences were revealed in relative abundance of *Azospira*, *Brevundimonas*, *Roseomonas*,

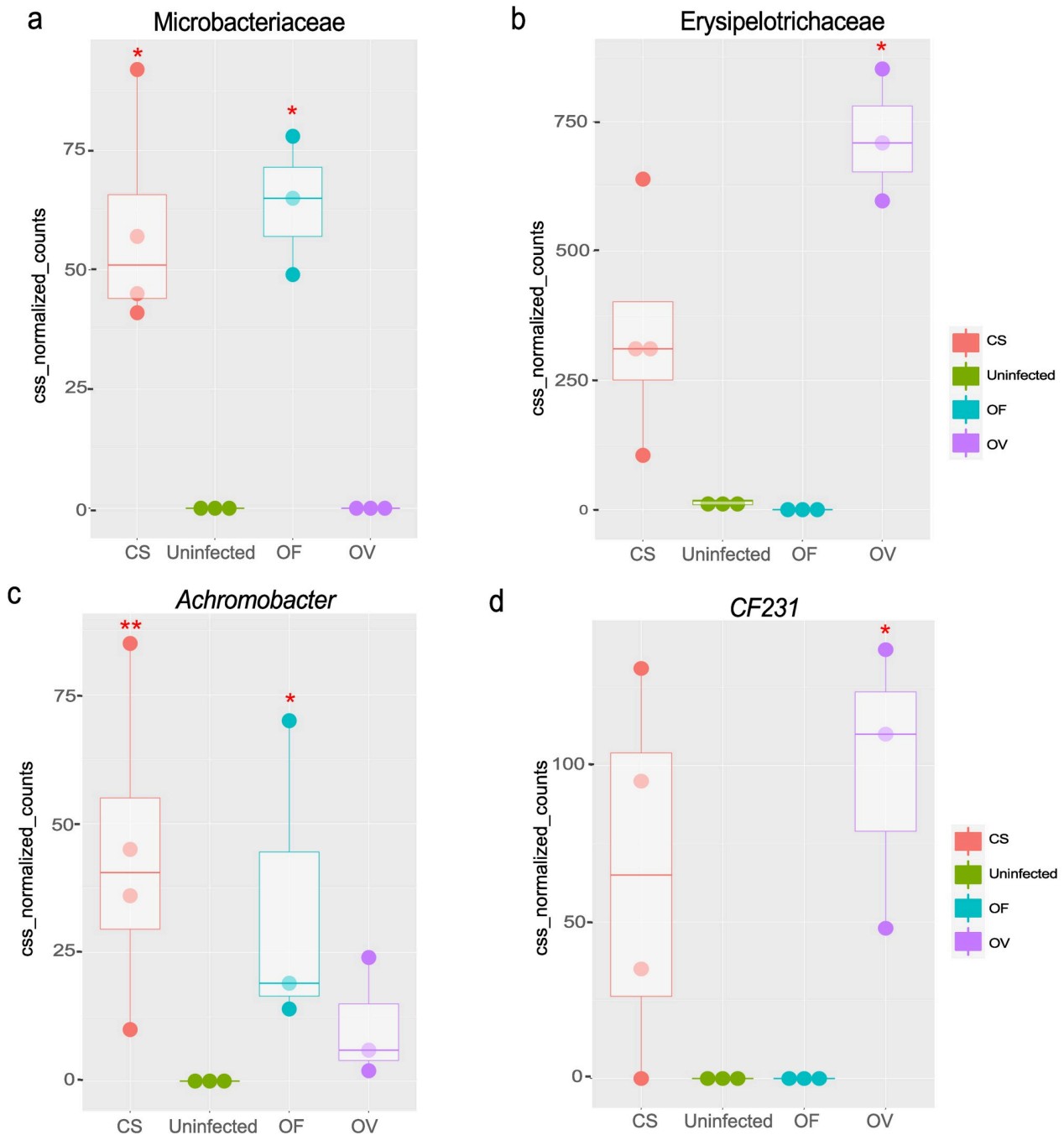

**Fig 5. Differently distributed taxa among bile samples.** Kruskal Wallis test followed by post-hoc Dunn's test of CSS-normalized counts with the Bonferroni correction. $^*P_{adj} < 0.05$ as compared to uninfected animals. OF: *O. felineus*, CS: *C. sinensis*, OV: *O. viverrini*.

*Achromobacter* (Fig 5C), *CF231* (Fig 5D), *Ralstonia* and others. For instance, a significant increase of Erysipelotrichaceae (Figs 5D), [Paraprevotellaceae], Acetobacteraceae, Coriobacteraceae and Corynebacteriaceae relative abundance was found in the *O. viverrini*–infected bile samples, whereas increased Microbacteriaceae and Enterobacteriaceae relative abundance was revealed in *C. sinensis*-infected bile (Fig 5A) ($P_{adj} < 0.05$; Table I in S1 Tables). *Achromobacter* relative abundance was significantly elevated in the *O. felineus* and *C. sinensis*-infected

bile (Fig 5A), whereas *CF231* and *Ralstonia* was significantly elevated only in the *O. viverrini*–infected bile samples (Fig 5D).

### 3.4. Colon fecal microbiomes

After removal of low-abundance features (1–2 counts), 1188 features were identified in the fecal samples. Taxonomic composition was as follows: the bacterial superkingdom, 9 phyla, 18 classes, 50 families, and 55 genera. In terms of taxonomy (observed features), fecal microbial communities were the least diverse of all sample types classified by the source of DNA (worms, feces, or bile) (Fig 1A).

Analysis of the index of phylogenetic diversity (Faith's PD, P value = 0.039) uncovered a significant increase within the colon fecal microbiome upon infection (Fig 6A), while Shannon's index showed no significant differences among trematode species (Fig 6B). According to the taxonomic analysis, 32 genera and 34 families were present in the microbiome of the feces of uninfected animals, while the total numbers in the infected animals were 55 genera and 50 families.

According to the Bray–Curtis distance metrics at the feature level (Fig 6C) *C. sinensis* and *O. felineus* infections significantly affected the composition of the fecal microbiome (pairwise permutation analysis of variance, q-value = 0.0465 each), whereas *O. viverrini* infection did not.

The top five most abundant families in uninfected and infected fecal samples were S24-7, Ruminococcaceae, Lachnospiraceae, Bacteroidaceae, and [Paraprevotellaceae], with relative abundance levels of 56–63%, 16–19%, 5–10%, 2.5–3.4%, and 1.0–2.4%, respectively.

Meanwhile, the top four most abundant genera in uninfected fecal samples were *Ruminococcus*, *Bacteroides*, *Oscillospira*, and *CF231*, with relative abundance levels of 39%, 14%, 10%, and 9.8%, respectively. After the infection, the top four most abundant genera in fecal samples were *Ruminococcus*, *Oscillospira*, *Bacteroides*, and *Allobaculum*, with relative abundance levels of 36.2%, 11.9%, 9.4%, and 7.5%, respectively.

ZigFit analysis revealed 54 features differentially present ($P_{adj} < 0.05$) in the infected colon fecal samples compared to uninfected ones, regardless of trematode species. Relative abundance of 28 of these decreased significantly ($\log_2$ fold change $< -3.6$) and that of 26 significantly increased ($\log_2$ fold change $> 3.5$) (Table J in S1 Tables). The presence-absence test indicated that nine features are significantly differentially present in the infected fecal samples compared to uninfected ones ($P_{adj} < 0.05$; Table K in S1 Tables).

Notably, significant differences were observed in the fecal microbiome depending on the species of infecting helminths; namely, *O. felineus* infection significantly changed 78 features ($P_{adj} < 0.05$, with 24 features downregulated and 54 upregulated) (Table L in S1 Tables), and *C. sinensis* infection changed the largest number of features: 295 ($P_{adj} < 0.05$, with 120 features downregulated and 175 upregulated, Table M in S1 Tables). *O. viverrini* infection significantly changed 219 features ($P_{adj} < 0.05$, with 99 features downregulated and 120 upregulated, Table N in S1 Tables).

There were 8 differentially distributed taxa in the Kruskal Wallis test followed by post-hoc Dunn's test on the genus level among feces samples, 11 taxa on the family level, and one taxa on the phylum level ($P_{adj} < 0.05$) (Table O in S1 Tables). The most pronounced changes were observed after the *C. sinensis* infection. For instance, this infection changed the amounts of *Streptococcus*, *Roseburia*, *Mucispirillum*, *Lactobacillus*, *Butyricimonas*, *AF12*, and *Acinetobacter* ($P_{adj} < 0.05$). *O. viverrini* infection altered relative abundance of *Sphingomonas*, *Roseburia*, and *Butyricimonas*. *O. felineus* infection changed the amounts of *Roseburia*, and *Butyricimonas* genera ($P_{adj} < 0.05$) (Table O in S1 Tables).

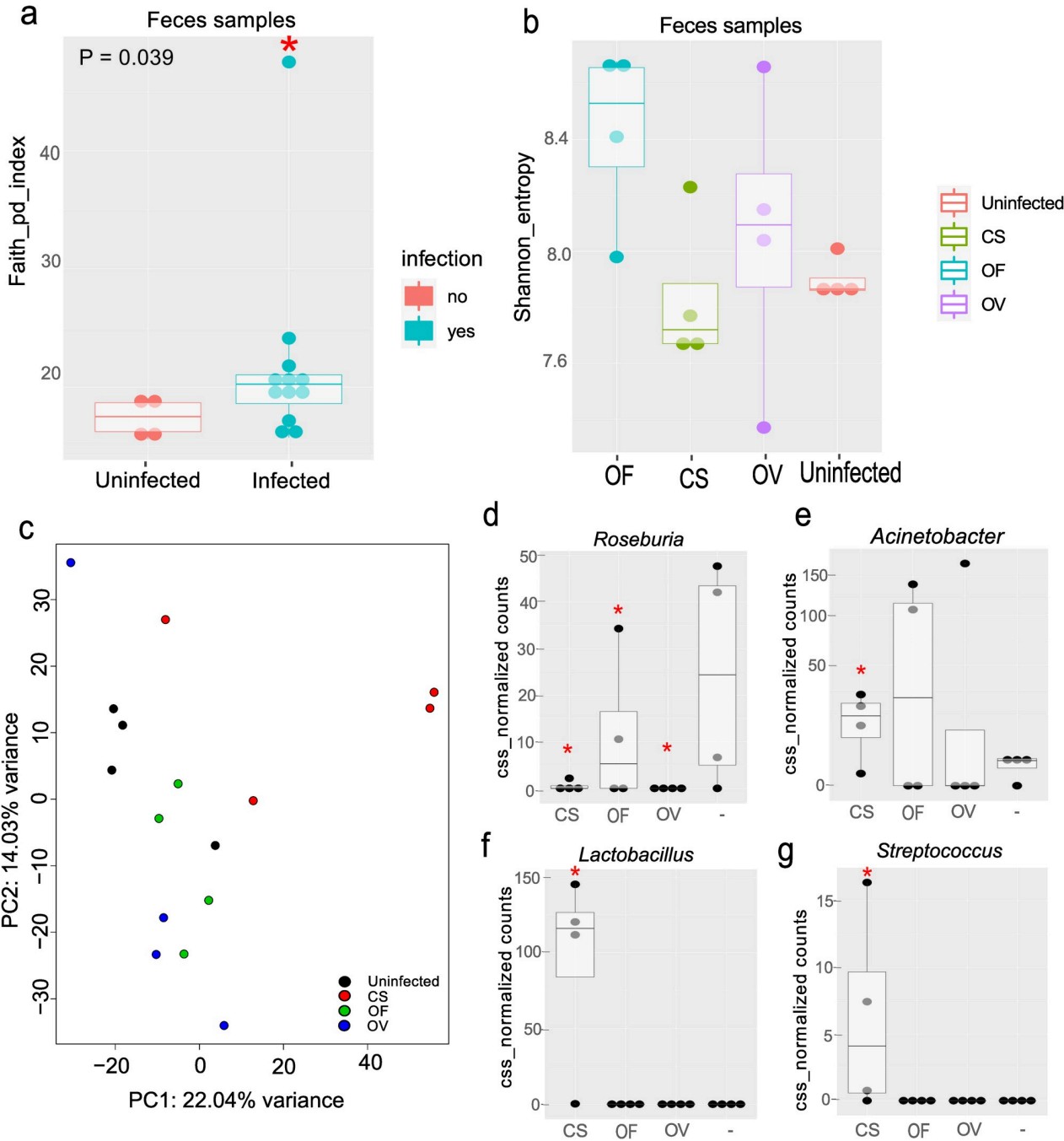

**Fig 6. The results of the alpha and beta diversity analysis of the fecal samples.** a. Differences in Faith's PD between uninfected and infected samples of feces. b. Shannon's index. c. PCoA based on Bray–Curtis distance metrics at the feature level of fecal samples from uninfected and infected hamsters (P = 0.014). d–g. Differently distributed genera among fecal samples. * False Discovery Rate < 0.05, significant changes as compared to the uninfected samples (Wilcoxon's test of CSS-normalized counts with the Benjamini–Hochberg correction). OF: *O. felineus*, CS: *C. sinensis*, OV: *O. viverrini*.

At the family level, in the gut microbiome after *C. sinensis* infection, significant alterations were observed in relative abundance of Streptococcaceae, Rikenellaceae, Porphyromonadaceae, Paraprevotellaceae, Odoribacteriaceae, Moraxellaceae, Lactobacillaceae, Deferribacteraceae and Coriobacteriaceae. *O. viverrini* infection changed relative abundance of three

families: Sphingomonadaceae, Odoribacteriaceae, and Coriobacteriaceae. *O. felineus* infection altered relative abundance of the following families: Rikenellaceae, Porphyromonadaceae, Coriobacteriaceae and Odoribacteriaceae ($P_{adj} < 0.05$) (Table O in S1 Tables).

Thus, *C. sinensis*, *O. felineus*, and *O. viverrini* each significantly reduced the number of commensal bacterial cells, e.g., Roseburia, in the gut microbiota of the host (Fig 6D). Additionally, *C. sinensis* infection significantly lowered the number of bacterial cells of genera *AF12*, recognized as beneficial. At the same time, the presence of such bacteria as *Acinetobacter* (Fig 6E), *Lactobacillus* (Fig 6F), and *Streptococcus* (Fig 6G), which cause severe or opportunistic infections, significantly increased. Overall, although infection with any of the three parasitic worm species induced significant changes in the gut microbiome (54 differentially present features; ZigFit, $P_{adj} < 0.05$), *C. sinensis* infection had the greatest impact on taxonomic composition, altering 295 of 500 features (Table M in S1 Tables).

## 3.5. Worm microbiomes

After removal of low-abundance features (1–2 counts), 989 features were identified in the worm samples. The following taxonomic composition was registered: the bacterial super-kingdom, 15 different phyla, 30 classes, 91 families, and 135 genera. The presence of Enterobacteriaceae as a major component of the worm microbiota (up to 4%) proved to be a main difference between the worm and bile microbiomes.

According to alpha and beta diversity indices, there were differences among three liver fluke microbiomes. The analysis uncovered a significant difference among worm group samples judging by the unweighted (phylogenetic) diversity index (Faith's PD; Table 1). *O. felineus* worms were the most phylogenetically diverse group of samples, whereas *C. sinensis* worm samples were the least diverse and differed significantly from the *O. felineus* (q-value = 0.003) and *O. viverrini* (q-value = 0.031) worm samples. Fisher's index was significantly higher in *O. felineus* (q-value = 0.019) and *O. viverrini* (q-value = 0.018) samples compared to *C. sinensis* samples (Table 1). There was no significant difference among the worm species in Shannon's index, which denotes weighted bacterial diversity and richness. Nonetheless, alpha diversity rarefaction curves for all samples reached plateaus, ***meaning*** that no further sequencing was needed (S6 Fig).

According to the weighed UniFrac analysis, the microbiome of *C. sinensis* significantly differed from the microbiomes of *O. viverrini* and *O. felineus* (q-value = 0.0465 each); however, no such difference was detected between the microbiomes of *O. viverrini* and *O. felineus* (Fig 7A). Similar results were obtained during the evaluation of beta diversity using Bray–Curtis and Jaccard distances.

## 3.6. Common and distinct bacterial taxa in the worm groups

The top five most abundant bacterial families in all worms were common, in particular, Sphingomonadaceae, S24-7, Chitinophagaceae, Enterobacteriaceae, and Bradyrhizobiaceae.

**Table 1. Alpha diversity indices of worm microbiomes.**

|  | *C. sinensis* | *O. felineus* | *O. viverrini* |
|---|---|---|---|
| **Shannon's index** | 7.5±0.4 | 7.4±0.1 | 7.8±0.1 |
| **Fisher's index** | 149.5±20.8 | 224.7±42.0 * | 226.0±5* |
| **Faith's PD** | 65.5±1.4 | 99.5±15.4 ** | 88.1±2.6 * |

*Significant differences from *C. sinensis* samples ($P_{adj} < 0.05$, ANOVA and Tukey's *post hoc* test).

Meanwhile, the top five most abundant genera in worm microbiomes were common, in particular, *Sphingomonas*, *Sediminibacterium*, *Ruminococcus*, *Allobaculum*, and *Rhodococcus*, with relative abundance levels of 18.4–20.1%, 9.0–9.8%, 1.9–2.6%, 3.7–4.0%, and 1.9–2.6%, respectively. All other genera of bacteria were present at less than 2%.

To analyze distinct representatives of the trematode microbiota, we excluded those species that were found in bile. Furthermore, we excluded minor taxa (1–2 counts and present in only one sample). Eventually, 77 features were identified for *O. viverrini*, 72 features for *O. felineus*, and 66 features for *C. sinensis* (Table P in S1 Tables). There were 25 features common among all the three liver fluke species. They included *Flavobacterium* spp., *Halomonas nitritophilus*, *Microbispora rosea*, and *Rhodoferax* spp. In addition, unique features for each trematode species were found too, namely, 24 features for *O. felineus*, 29 features for *O. viverrini*, and 29 features for *C. sinensis*.

It is noteworthy that bacteria that are mesophilic or thermophilic found in soil were also identified, such as *Microbispora* or *Halomonas*, *Deinococcus*, and *Nostoc* spp. *Nostoc* species, which are nitrogen-fixing cyanobacteria, are fairly widespread and have been found in a variety of environmental niches in soil and at the bottom of freshwater bodies. Representatives of Gemmatimonadetes (only in *O. viverrini*) were also registered in our study; they are usually detected in the soil microbiomes. Notably, not only soil- and water-related species but also fish pathogens were registered here, such as Piscirickettsiaceae spp (Table P in S1 Tables).

Pathogenic bacteria were identified in the worm microbiomes as well, in particular, various *Campylobacter* spp. For instance, *Campylobacter rectus* was found in *C. sinensis*, and *Campylobacter ureolyticus* in *O. viverrini*. No *Helicobacter* species were found. *Staphylococcus aureus* was found in *C. sinensis* and *O. viverrini* helminths. *Trueperella* spp. was registered in *O. felineus* and in *O. viverrini* worms. Several species of the Enterobacteriaceae family were found, in particular *Klebsiella*, *Salmonella*, *Providencia*, and *Trabulsiella* spp. The last three were detectable only in *O. viverrini*. *Salmonella enterica* was found in *O. felineus* and *C. sinensis* worms.

In the MetagenomeSeq analysis, 114 features were documented with different distributions between *C. sinensis* and *O. viverrini* (Table P in S1 Tables21), 132 features with different distributions between groups *C. sinensis* and *O. felineus* (Table R in S1 Tables). It is noteworthy that the smallest difference in bacterial abundance was noted between *O. viverrini* and *O. felineus* (40 features) (Table S in S1 Tables; Table T in S1 Tables).

## 3.7. Enterobacteriaceae in hamster liver tissue

Relative abundance of the Enterobacteriaceae family significantly went up in the bile of animals after the infection with any of the three liver fluke species. Additionally, both commensal and pathogenic genera, such as *Salmonella*, *Gluconacetobacter*, and others, were found among the species of this family. It was important for us to corroborate the significant upsurge of bacteria from this family in the liver, as confirmation for the results of statistical analysis of microbiome sequencing. Accordingly, by immunohistochemistry, we examined the liver of infected animals for the presence of a specific signal of Lipid A from the Enterobacteriaceae family (Fig 8). Thus, it was noted that in the liver of uninfected animals, bacterial cells are undetectable (Fig 8A). In contrast, after infection, both inside large bile ducts and inside small ones, a specific signal of these bacteria was detected (Fig 8B–8F). Besides, these bacteria were found inside the worm gut (Fig 8C), consistently with our sequencing finding that 4% of the total worm microbiota was represented by this family.

It is noteworthy that at sites of bile duct epithelium injury (Fig 8E), bacteria were seen entering the periductal area of the liver. The same figure (Fig 8E) shows that the epithelium is not a single layer of cells as in the control (Fig 8A) but is arranged in multiple layers; this

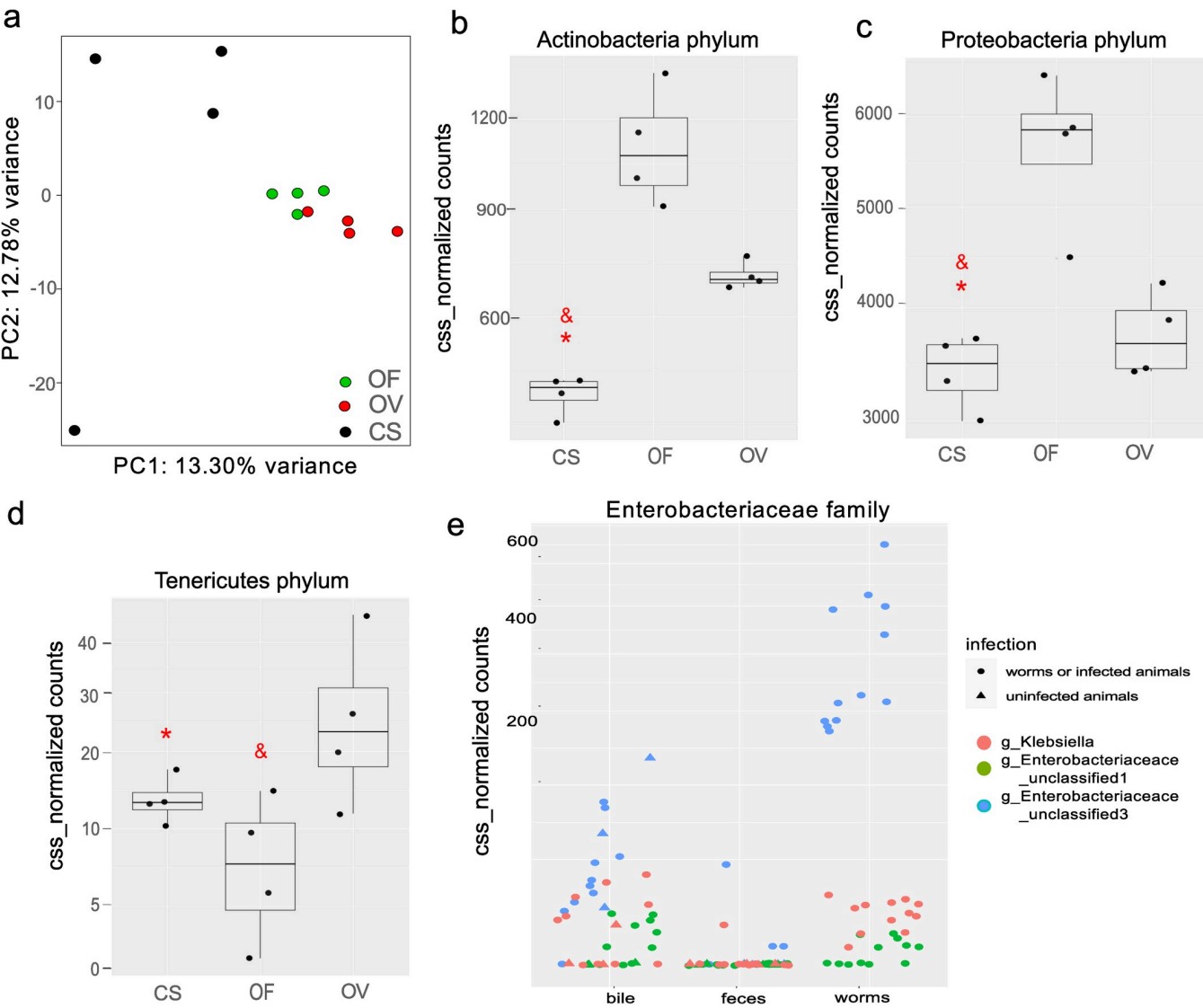

**Fig 7. Results of alpha and beta diversity analyses of the worm samples.** a. Beta diversity. Weighed UniFrac analysis (pseudo-F = 1.3613, P = 0.002). b–d. Differently distributed phyla among worm samples. e. Worm samples are enriched with Enterobacteriaceae bacteria. * False Discovery Rate < 0.05, significant differences from the *O. felineus* samples, &False Discovery Rate < 0.05, significant differences from the *O. viverrini* samples (Wilcoxon's test of CSS-normalized counts with the Benjamini–Hochberg correction).

problem is probably accompanied by a modification of intercellular contacts (Fig 8E and 8F) and facilitates the penetration of bacteria (Fig 8E).

## Discussion

In this study, we demonstrated that related liver flukes *O. felineus*, *O. viverrini*, and *C. sinensis* under standardized experimental conditions cause both similar and species-specific qualitative and quantitative changes in microbiota composition of bile and feces in experimental hamsters infected with these trematodes. These alterations primarily affect relative abundance of individual features and phylogenetic diversity of microbiomes of the infected hamsters and are trematode-specific. It is worth mentioning that 12 features changed in bile regardless of the parasite species. In contrast, in bile, 92 features ($P_{adj} < 0.05$) changed after *O. viverrini*

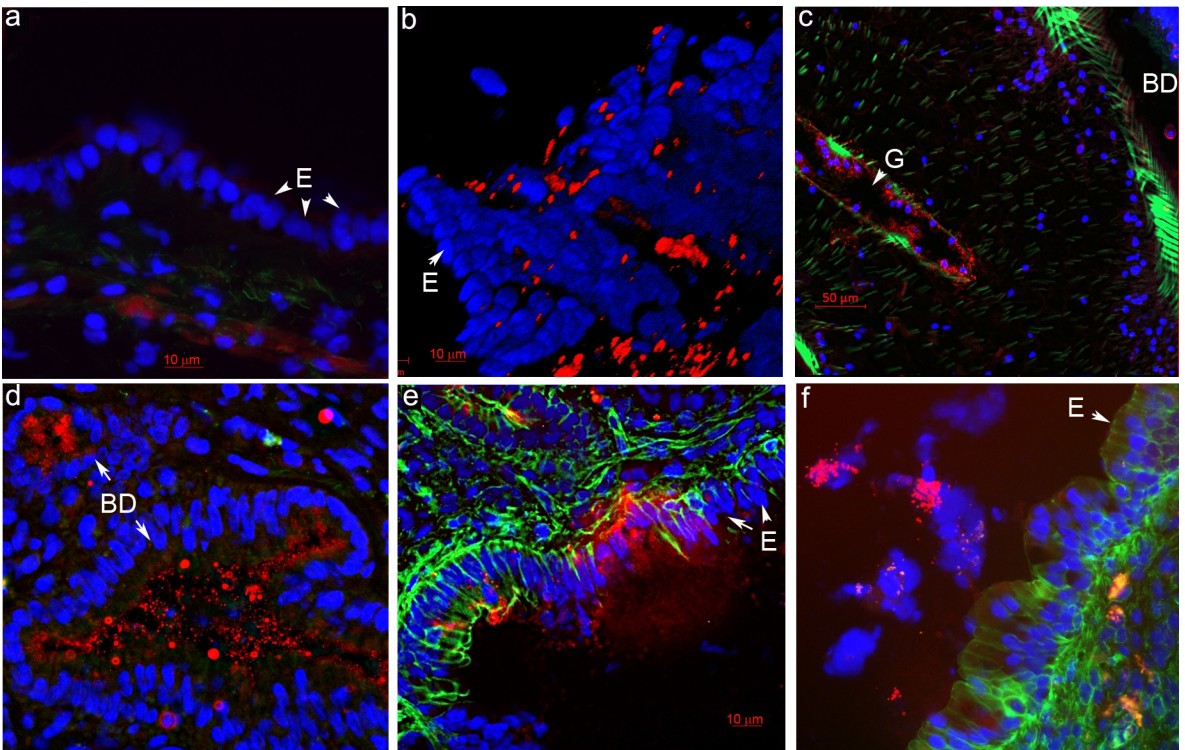

**Fig 8. Enterobacteriaceae (Lipid A) in the liver of the uninfected and the liver fluke-infected hamsters.** a. An uninfected hamster. b. 3D reconstruction of the internal surface of a bile duct after confocal microscopy reveals the presence of Enterobacteriaceae Lipid A. c. Bacteria inside the gut of the *O. viverrini* parasite. d. Enterobacteriaceae presence inside small bile ducts of an *O. viverrini*–infected hamster. e. Penetration of bacteria through the injured epithelium in the bile duct of an *O. felineus*–infected hamster. f. A multilayered epithelium in the bile duct of a *C. sinensis*–infected hamster. E: epithelial cells; BD: bile duct; red color: Lipid A of Enterobacteriaceae; green color: actin filaments (Phalloidin 488 staining); blue color: nuclei (DAPI staining). E: epithelium of bile duct; BD: bile duct; G: gut of a worm.

infection, 55 features changed after *C. sinensis* infection, and 36 features changed after *O. felineus* infection. Of note, *O. viverrini*, a fluke with the most pronounced carcinogenic potential, has the strongest impact on qualitative composition of the bile microbiome. *O. felineus*, which is not a biological carcinogen [8], caused the smallest changes in the bile microbiome; *C. sinensis* had a moderate effect on the bile microbiome.

Notably, the bile phylogenetic diversity index in *O. viverrini*–infected hamsters proved to be significantly higher than that in the uninfected animals, in agreement with previously published data on *O. viverrini*–infected hamsters [24–25] and on opisthorchiasis patients [26]. In particular, we observed significantly higher relative abundance of Bifidobacteriaceae, Erysipelotrichaceae, [Paraprevotellaceae], Acetobacteraceae, Coriobacteraceae and Corynebacteriaceae and of such bacterial genera as *Roseomonas*, *CF231* and *Ralstonia* in *O. viverrini*–infected bile samples. Increased abundance of some of these bacteria are associated with gastrointestinal inflammation in mice [27–28] or in patients with HCC-cirrhosis, inflammatory bowel diseases, colorectal cancer patients [29–31]. Erysipelotrichaceae spp. appear to be highly immunogenic [32]. Moreover, the relative abundance of Erysipelotrichi positively correlated with tumor necrosis factor alpha levels [33].

Reports have indicated that an imbalance of commensals [31, 34, 35] as well as presence of certain pathogenic bacteria, can promote various types of cancers. An overgrowth of Bifidobacteriaceae and Coriobacteriaceae spp. commensals, that were upregulated only in *O. viverrini*–infected bile, can influence the pathogenesis of bile duct disorders through bile acid

metabolism, because they are involved in oxidation and epimerization of bile acids [36–37]. Secondary bile acids can trigger oxidative damage. On the other hand, increased abundance of commensals may cause oxidative stress and oxidative damage of DNA in epithelial cells because various anaerobic fermentative bacteria belonging to [Paraprevotellaceae] and Coriobacteriaceae spp. generate ethanol [36].

Thus, the data suggest that *O. viverrini* alters the bile microbiome the most, and in addition, bacteria associated with inflammation become significantly elevated.

In contrast to this, *O. felineus* and *C. sinensis* infection significantly less alter the bile microbiota. Nevertheless, some inflammation-associated bacteria or opportunistic pathogens have also been found. In particular, upregulation of *Achromobacter*, *Microbacterium* and *Micrococcus* relative abundances in bile were characteristic for *O. felineus* and *C. sinensis* infection, *Klebsiella*–for *C sinensis* infection. Gastrointestinal colonization of *Klebsiella* may precede abdominal infections [38]. *Achromobacter* and *Micrococcus* are opportunistic pathogens that become virulent with immunocompromised and unhealthy individuals in certain conditions, such as cystic fibrosis, hematologic and solid organ malignancies, renal failure, and certain immune deficiencies [39–40]. Although an increase in *H. pylori* abundance has been previously demonstrated by *UreA* gene-specific PCR in liver fluke infected SPF hamsters [7], we did not show the presence of *Helicobacter spp.* in this study. This research was based on amplification of the 16S ribosomal RNA gene using non-specific bacterial primers. In order to maintain as much bacterial diversity as possible, the number of PCR cycles was kept to a minimum (25 cycles for bile, 20 cycles for feces). Unfortunately, minimal number of PCR cycles was not enough to identify all low-presented bacterial species. The results of this study suggest that *H. pylori* was not a major species.

It is noteworthy that relative abundance of the Enterobacteriaceae family significantly went up in the bile of animals after the infection with any of the three liver fluke species. Members of this family may be responsible for epithelial damage by producing genotoxic metabolite colibactin. Colibactin produced by the polyketide synthase was found in genomes of some representatives of Enterobacteriaceae. It forms DNA inter-strand cross-links and causes mutations leading to colon and biliary tract cancer [35, 41].

Higher abundance of Bifidobacteriaceae, Enterobacteriaceae, and Enterococcaceae bacteria has been documented in people with opisthorchiasis viverrini complicated by CCA in bile as compared to patients with CCA without *O. viverrini* infection [42]. In the bile of patients with cholelithiasis aggravated by *O. felineus* infection, greater abundance of such bacteria as *Klebsiella*, *Aggregatibacter*, *Lactobacillus*, and *Haemophilus* is observed, as compared with patients not infected with *O. felineus* [43].

What is the reason for bile microbiota changes after the liver fluke infections? One of possible explanations is the induction or suppression of physiological activities of various bacteria owing to the emergence of inflammatory foci in the biliary tract under the influence of trematode infection. Indeed, we demonstrated that Enterobacteriaceae—present in trace amounts in the uninfected animals—actively expand after the infection and colonize the biliary epithelium. In addition, as a consequence of the liver fluke infection, an imbalance in the metabolic composition of bile develops. This in turn disturbs functions of the intestinal barrier and the entry of the intestinal microbiota and its products into the liver [44]. Therefore, a circle of mutually inducing pathological processes can arise leading to qualitative and quantitative rearrangements of microbiota composition not only of bile but also of intestines [44].

In support of this notion, we registered marked alterations in the composition of the fecal microbiota after the fluke infections of the hamsters, and similarly to the bile microbiome, there were similar and species-specific changes. The smallest modification of the fecal microbiome occurred after *O. felineus* infection: relative abundance of 78 features

changed ($P_{adj} < 0.05$), consistently with the effect of *O. felineus* on the bile microbiome. In contrast, *C. sinensis* infection had the most pronounced effect on the colon fecal microbiota: the abundance of 295 features changed significantly ($P_{adj} < 0.05$); after *O. viverrini* infection, a large number of alterations in the fecal microbiome were registered too: 219 features were affected ($P_{adj} < 0.05$).

Of note, during the rearrangement of the fecal microbiome, relative abundance of bacteria associated with pathologies went up, and accordingly, relative abundance of bacteria associated with health diminished. For instance, the emergence of such opportunistic pathogens as *Acinetobacter*, *Pseudomonas*, and *Streptococcus*, which were absent in the uninfected hamsters, was demonstrated after *C. sinensis* infection. On the other hand, relative abundance of such genera as *Parabacteroides*, *Roseburia*, and *AF-12* significantly decreased. Genera *Parabacteroides* and *Roseburia* are common commensals in the microbiota of the mammalian colon [45]; *AF12* spp. have been found to be beneficial [46–47]. Moreover, all three species of liver flukes raised Enterobacteriaceae abundance and lowered *Roseburia* abundance; this pattern is typical for many human inflammatory diseases [45, 48–50]. Our findings are in line with the data indicating significant taxonomic changes in *F. hepatica*–positive ruminants [51] and in *O. viverrini*–infected patients [4].

Similar changes in the intestinal microbiota have been registered in some chronic diseases of the gastrointestinal tract [52]. Moreover, some gut microbiota species like *Fusobacterium* spp. are reported to contribute to colorectal cancer [6]. The nematode *Trichuris suis* has been found to significantly modify the pig colon microbiota: a reduction in relative abundance of Fibrobacter, Parabacteroides, and Ruminococcus and exacerbation of campylobacteriosis [53]. Some research findings point to a decrease in the amount of Firmicutes bacteria (e.g., *Clostridium leptum* and *Faecalibacterium prausnitzii*) and an increase in Proteobacteria abundance in chronically infected individuals [54–55]. Furthermore, higher abundance of families Lactobacillaceae and Enterobacteriaceae and lower abundance of genera *Eubacterium* and *Clostridium* have been demonstrated in mice infected with the helminth *Heligmosomoides polygyrus* [56–57]. Infection with the nematode *Trichuris muris* in mice raises Lactobacillaceae abundance in the ileum [58]. In our study, we also found an increase in Lactobacillus relative abundance in the colon feces of *C. sinensis*–infected hamsters. At the same time, these bacteria were not detectable in the intestine of the uninfected hamsters.

Thus, liver fluke infections have a systemic influence on the microbiota of the biliary tract and intestine, resulting in upregulation of potentially pathogenic features. Is this only a result of uncontrolled growth of host bacteria under conditions of homeostasis disturbed by infection, or does the helminth microbiome also contribute to the total number of changes in the bile and feces microbiotas?

In our work, among the bacteria that were found only in helminths and were absent in other types of samples, some genera were found that are characteristic of external environments including soils, sewage, freshwater bodies, and sludge. The worms were found to contain unique classes of bacteria (Epsilonproteobacteria, Gemmatimonadetes, and Nostocophycideae). Some of the detectable genera belong to extremophiles, in particular, genera *Nostoc* and *Deinococcus* [59]. Here, microbial communities of adult trematodes differed significantly in alpha and beta diversity and also in taxonomic diversity at different taxonomic levels. For example, in terms of the number of features as well as phylogenetic diversity (Faith's PD), the microbiome of *C. sinensis* helminths proved to be significantly poorer than that of the other two liver flukes, although frequencies of features were the same among all the three species. Meanwhile, phylogenetic diversity of the *O. felineus* microbiome (Faith's PD = 99.5) was the highest in comparison to *C. sinensis* and *O. viverrini* worms (Faith's PD = 65.5 and 88.5,

respectively). Gemmatimonadetes bacteria were found only in *O. viverrini* worms, while Gluconacetobacter and Serratia were registered only in the *O. felineus* microbiome.

At the same time, only *O. viverrini* infection (Shannon's index) significantly affected the alpha diversity of bile by changing the largest number of features. Thus, the diversity of the helminth microbiome itself is not a determinant of the host bile microbiome. Of note, the hypothesis that a microbiota is transferred from a parasite to a mammalian host is currently quite popular as the main reason explaining the complicated morbidity of helminthiases.

Numerous published data support this hypothesis. For instance, the transfer of the alpha-proteobacterium *Wolbachia pipientis* from one host to another during the life cycle is observed in nematodes *Dirofilaria immitis*, *D. repens*, and *Onchocerca volvulus* [60–62]. Furthermore, the nematode *Heterakis gallinarum* is reported to be a reservoir for the protozoan *Histomonas meleagridis*, which causes the disease histomosis (black head) in turkeys [63]. The trematode *Nanophyetus salmincola* is a carrier of the alpha-proteobacterium *Neorickettsia helminthoeca*, which is the causative agent of salmon poisoning disease in dogs [64]. The fluke *Acanthatrium oregonense* (and some others), when entering the body of a horse, infects it with *Neorickettsia risticii*, causing Potomac horse fever [65].

We also showed that many infection-associated bacterial species found in the infected bile also exist in helminths, such as *Haemophilus*, *Klebsiella*, *Finegoldia*, *Methyloversatilis*, and *Deinococcus* spp. The presence of these bacteria both in bile after the infection and in the helminth microbiota can probably be attributed to the fact that these bacteria are transferred to the host from trematodes.

In the infected bile, however, we also detected many other bacteria that were not found inside the helminths, and therefore their presence cannot be explained by the transfer from helminths to the host. These bacteria can be due to secondary bacterial infection. In particular, these bacteria include *Burkholderia*, *Phenylobacterium*, *Blastomonas*, and *Serratia* spp. One way or another, bacteria found after the infection probably contribute to the pathogenicity of diseases.

The microbiome differences in alpha and beta diversity observed here among the three liver flukes were expected. On the one hand, all three species have territorial disunity, are endemic for areas with a distinct climate, and have different species of the first intermediate hosts (freshwater snails). In addition, the presence of two intermediate aquatic hosts in the life cycle and of one free-living aquatic stage led us to expect that all three trematode species collected in different regions of the world would have different microbiomes. Unexpectedly, major members of the microbiome turned out to be the same among the three species of trematodes while also coinciding with major representatives of the bile microbiome in the uninfected hamsters. Consequently, the known hypothesis that trematodes act as a microbiota reservoir and infect the mammalian host can be supplemented by a reverse phenomenon: a large mammalian host infects the small parasitic organism with its microbiota, thereby ensuring a mutual exchange of microorganisms.

## Conclusion

In this study, we compared microbiomes of three liver flukes for the first time in the same model under the same conditions in experimental SPF hamsters. We showed that the microbiome of each liver fluke (closely related opisthorchiids *O. felineus*, *O. viverrini*, and *C. sinensis*) contains its own unique bacteria and contributes to significant species-specific alterations in microbiomes of the bile and intestinal content of the host. *O. viverrini* had the most potent effect on the host bile microbiome, whereas *O. felineus* had the smallest impact on host bile microbiome. These findings are in line with data on unequal carcinogenic potential of

opisthorchiids. Relative abundance of bacteria associated with gastrointestinal inflammation (Bifidobacteriaceae, Erysipelotrichaceae, and of such bacterial genera as *Roseomonas*, *CF231* and *Ralstonia*) in *O. viverrini*–infected bile was significantly elevated. Upregulation of *Achromobacter*, *Microbacterium* and *Micrococcus* relative abundances in host bile were characteristic for *O. felineus* and *C. sinensis* infection. All three infections significantly increased Enterobacteriaceae abundance in host bile.

*C. sinensis* infection had the most pronounced effect on the colon fecal microbiota: the abundance of 295 features was significantly modified increasing the abundance of potential opportunistic pathogens, including *Acinetobacter*, *Streptococcus* and Lactobacillaceae. *O. felineus* and *O. viverrini* had smaller impact on host colon microbiome.

Analysis of the three-way relationship among the mammalian host, microbiomes, and trematodes can lead either directly to the development of novel methods for parasite control or indirectly (through a more accurate understanding of the pathogenesis) to the design of more effective modalities for diagnosis, prevention, and treatment.

## Supporting information

**S1 Fig. Real-time PCR results (2nd round of amplification) for amplicons prepared from DNA samples isolated from feces.**
(TIF)

**S2 Fig. Results of High Sensitivity DNA Assay (Agilent 2100) of DNA libraries constructed.**
(TIF)

**S3 Fig. Number of samples used for construction of DNA libraries.**
(TIF)

**S4 Fig. Output statistics of raw sequencing data.**
(TIF)

**S5 Fig. Most abundant worm and bile microbial communities.**
(TIF)

**S6 Fig. Alpha diversity Rarefaction Curve.** Alpha rarefaction parameter demonstrating species richness from the results of sampling.
(TIF)

**S1 Tables.**  Table A in S1 Tables. Pipeline employed for bioinformatics analysis. Table B in S1 Tables. DADA2 output statistics. Table C in S1 Tables. Statistics on identified features. Table D in S1 Tables. Features differentially presented in the infected bile samples compared to uninfected bile samples. Table E in S1 Tables. Presence-Absence test on bile samples (Infection vs Control). Table F in S1 Tables. Features differentially presented in OF bile versus control samples. Table G in S1 Tables. Features differentially presented in CS bile versus control samples. Table H in S1 Tables. Features differentially presented in OV bile versus control samples. Table I in S1 Tables. Differentially represented taxa among worm samples. Table J in S1 Tables. Features differentially presented in the infected fecal samples compared to uninfected fecal samples. Table K in S1 Tables. Presence-Absence test on feces samples (Infection vs Control) _Metagenomeseq. Table L in S1 Tables. Features differentially presented in OF feces versus control samples. Table M in S1 Tables. Features differentially presented in CS feces versus control samples. Table N in S1 Tables. Features differentially presented in OV feces versus control samples. Table O in S1 Tables. Differentially represented taxa in feces of hamsters.

Table P in S1 Tables. The list of unique species for worms only, not identified in any other samples. Table Q in S1 Tables. Features differentially presented in OV worm microbiomes versus CS worm microbiomes. Table R in S1 Tables. Features differentially presented in CS worm microbiomes versus OF worm microbiomes. Table S in S1 Tables. Differentially represented taxa among worm samples. Table T in S1 Tables. Features differentially presented in OV worm microbiomes versus OF worm microbiomes.
(XLSX)

## Acknowledgments

We are thankful to Dr. Natalia Bondar for the assistance in DNA library preparation. The English language was corrected by shevchuk-editing.com.

## Author Contributions

**Conceptualization:** Sung-Jong Hong, Banchob Sripa, Viatcheslav A. Mordvinov.

**Data curation:** Ekaterina A. Lishai, Oxana Zaparina, Nina V. Baginskaya.

**Formal analysis:** Maria Y. Pakharukova, Ekaterina A. Lishai.

**Funding acquisition:** Maria Y. Pakharukova.

**Investigation:** Maria Y. Pakharukova, Ekaterina A. Lishai, Oxana Zaparina, Nina V. Baginskaya.

**Methodology:** Ekaterina A. Lishai, Oxana Zaparina, Nina V. Baginskaya, Banchob Sripa.

**Project administration:** Maria Y. Pakharukova, Viatcheslav A. Mordvinov.

**Resources:** Sung-Jong Hong, Banchob Sripa, Viatcheslav A. Mordvinov.

**Supervision:** Maria Y. Pakharukova, Sung-Jong Hong, Banchob Sripa, Viatcheslav A. Mordvinov.

**Visualization:** Maria Y. Pakharukova, Nina V. Baginskaya.

**Writing – original draft:** Maria Y. Pakharukova.

**Writing – review & editing:** Maria Y. Pakharukova, Ekaterina A. Lishai, Oxana Zaparina, Sung-Jong Hong, Viatcheslav A. Mordvinov.

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
