## [Decision Letter · Decision Letter 0]

7 Oct 2022

Dear Mariya Y Pakharukova,

Thank you very much for submitting your manuscript "Opisthorchis viverrini, Clonorchis sinensis and Opisthorchis felineus liver flukes affect mammalian host microbiome in a species-specific manner" for consideration at PLOS Neglected Tropical Diseases. As with all papers reviewed by the journal, your manuscript was reviewed by members of the editorial board and by several independent reviewers. In light of the reviews (below this email), we would like to invite the resubmission of a significantly-revised version that takes into account the reviewers' comments. 

We cannot make any decision about publication until we have seen the revised manuscript and your response to the reviewers' comments. Your revised manuscript is also likely to be sent to reviewers for further evaluation.

Sincerely,

Wannaporn Ittiprasert, Ph.D

Academic Editor

Francesca Tamarozzi

Section Editor

Reviewer's Responses to Questions

**Key Review Criteria Required for Acceptance?**

**Methods**

-Are the objectives of the study clearly articulated with a clear testable hypothesis stated?

-Is the study design appropriate to address the stated objectives?

-Is the population clearly described and appropriate for the hypothesis being tested?

-Is the sample size sufficient to ensure adequate power to address the hypothesis being tested?

-Were correct statistical analysis used to support conclusions?

-Are there concerns about ethical or regulatory requirements being met?

Reviewer #1: The objective of this study is clearly address with clear hypothesis. However, in the method, due to the geographically different of each type of parasite, the authors did not provide several informations about the animal rearing process which might effect the microbiome such as location of hamster rearing, how the metacercaria was prepared and kept during the shipment if all group of hamsters was kept in the same location, or if the hamsters was kept in different locations, how the hamsters food was handling etc.

Reviewer #2: The manuscript entitled “Opisthorchis viverrini, Clonorchis sinensis and Opisthorchis felineus liver flukes affect mammalian host microbiome in a species-specific manner” by Pakharukova et al. showed that infections with these trematodes alter the composition of the microbiota in the bile and colon feces of experimental SPF animals and also examined the microbes of all liver flukes. There are several points concerning clarity in the manuscript. I think that the authors’ analysis needs to be reconsidered to make the most of the data they have collected. The author is unclaer to mention bioinformatics analysis in the method section, and the results are confusing. For example, OTU removal can have an effect on alpha diversity indices. Furthermore, how similar are the sequences assigned to the same OTUs, and what do you do for future annotation? The manuscript format must be improved for PNTD's style. I recommend a major revision.

Abstract 

- Lines 40-46; The authors highlight O. viverrini and C. sinensis in the result section, but no mention of O. felineus or the characteristics of their result.

Methods

- Line 132; In the SPF animals, authors should define what is free, whether it is bacterial free, helminth free, or both.

- Line 147; As I understand it, the other group is the uninfected group, which should also be mentioned in it.

- Line 157; Please check the format of the PNTD guide for the authors.

- Lines 164-165; According to the sentence "Pieces (10 to 15 mg) of frozen samples of individual adult worms were subjected to DNA extraction" How did the author process this? Did the author cut the worm's body? Do you utilize the whole worm body? What section of the worm's body was analyzed? And do all of the samples come from the same section? Because the quantity and species of bacteria in each organ vary, the author should clarify.

- Lines 173; Incomplete references to the author's supplementary information. Please double-check this error, which can be found throughout the article.

- Lines 183; As for negative control samples, what type of water was used? The author should provide details.

- The BioProject PRJNA866652 is not available online may be a reviewer link needs to be given for review.

Reviewer #4: Yes，it is.

**Results**

-Does the analysis presented match the analysis plan?

-Are the results clearly and completely presented?

-Are the figures (Tables, Images) of sufficient quality for clarity?

Reviewer #1: Most of the results and figures are clearly demonstrate. However, there are some sentence in the results part and discussion part which not agree with each other for example "No Helicobacter species were found" in Line 534 (Results) but in the discussion part the authors state that "On the other hand, no significant difference in Helicobacter abundance were observe..." (Line 706-707; discussion). The authors must carefully check about these points.

For the result presentation, the authors should considers the revise by avoiding the use of 'g_' 'f_' but using 'genus' 'family' following the name instead.

Reviewer #2: - Line 251; Author's features = Operational Taxonomic Units (OTUs) should also be specified. I strongly suggest authors provide supporting data indicating the abundance of each OTU (number of sequences) in each library. The authors must recognize that in this case, OTU abundance must be used to determine alpha diversity. The removal of OTU before or after taxonomic analysis, as well as before or after alpha diversity, should be clearly explained by the author.

- Lines 262-264; …“For instance, alpha diversity shows the richness of phylogeny. Beta diversity is calculated as the number of species that are not the same in two different groups.”…. In my opinion, this sentence does not have to be included in the results.

- Line 266; This is incorrect information. Fig1B is not Faith’s PD. Please check again. This, and other mistakes, may be found throughout the article.

- Line 281; I recommend writing text in sequence with the figure or rearranging the figure. It is recommended to do the same for all of the author's figures.

- Lines 286-292; The author should rewrite and point to the findings that the author aims to demonstrate. Please check the information since, as indicated in Figure 2, the worm specimen contains 16 phyla.

- Line 295; Figure 2. The author is not shown at the genus level.

- Line 297; Fig 2C, please add Y-axis name.

- Line 307; Throughout the text, but especially in Fig 2C, it is unclear how the authors combined experimental data from individuals. For example, using an average abundance. The methodology is not described.

- Line 317; Figure 3, I don’t think that Figure 3 contributes much to description of data. Maybe an alternative version of this figure or just description of data in main text.

- Line 351; the word “new genera” What exactly does this mean? It should be made clear. How is this new?

- Throughout the manuscript, I recommend reordering your Figures and Supplementary tables. Let's start with the numbers, in the order they were mentioned.

- Line 388; The result in Fig.5, it seems do not reach the level of statistical significance. I recommend removing all reporting of non-significant results.

- Line 395; As I suggested, OTU removal can have an effect on alpha diversity indices but can do after taxonomy. However, how similar are the sequences that were allocated to the same OTUs, and what do you do for further annotation? It's not mentioned in the method section.

- Line 413; There is conflicting information with the mentioned above, that there are 50 or 54 families.

- Line 415; I recommend removing Fig. 6C since it seems less important. Maybe an alternative version of this figure, for instance, by adding a figure of relative abundance (lines 420–428).

- Line 465; This is incorrect information. Fig6A is not matched with the text. Please check.

- Line 506; The author has written too freely for this sub-topic. Because the author has provided information about the outcomes in each sub-topic, I can deal with the main point. This sub-topic should provide the most important results that the reader needs to know.

- Line 570; The letters pointing in the image should be reshaped, such as E to e, to make it easier to see. Moreover, I see a duplicate sentence in Fig8 legend, please check.

Reviewer #4: Yes，It is.

**Conclusions**

-Are the conclusions supported by the data presented?

-Are the limitations of analysis clearly described?

-Do the authors discuss how these data can be helpful to advance our understanding of the topic under study?

-Is public health relevance addressed?

Reviewer #1: The authors should address more about the effect of the changes of each type of microbe on the pathology of biliary tract which currently less mentioning. Most of the discussion part is only compared the result with previous findings which not helpful with the understanding of the topic under study. 

The authors still not discuss about limitations of the study.

Reviewer #2: - The authors have introduced the importance of liver flukes in the development of cholangiocarcinoma. The author's question about the mechanisms underlying helminth-induced carcinogenesis with their microbes. Therefore, the authors should discuss the results obtained in the context of the bacteria found in the worm and in the host bile duct and should also discuss why this bacterium is found and how it is important or transmitted in the development of cholangiocarcinoma. For example, the results in Fig2, which, if possible, should be reached at the genus or species level to identify the function and effect of that back. Which species is related to inflammation or fibrosis? This is useful in predicting pathogenesis in the host bile duct or trying to find out which microbiome data has such an organism in your results.

- Lines 706-708; Did you find Helicobacter pylori in this study?

Reviewer #4: Yes,it is.

**Editorial and Data Presentation Modifications?**

Reviewer #1: This articles should be 'minor revision' due to the reason mentioning above.

Reviewer #2: No

Reviewer #4: Minor revision.

**Summary and General Comments**

Reviewer #1: Overall, the findings from the study is novelty and displays an important piece of knowledge in the study field. However, several part still need to modified especially on the discussion which will improve the understanding of the topic under study.

Reviewer #2: The author is unclear to mention bioinformatics analysis in the method section and the results are confusing. In particular, OTU removal can have an effect on alpha diversity indices. Furthermore, how similar are the sequences assigned to the same OTUs? Supporting data indicating the abundance of each OTU (number of sequences) in each library should be provided.

Reviewer #4: This manuscript showed us an interesting story that O. felineus, O. viverrini, and C. sinensis infections can induce changes in the composition of microbiota of bile and colon feces of experimental animals infected with these trematodes. However, we have some comments:

1. Most results are the single description of the sequence result, i think it is better to add some comparion of the different micribiome of the three orgin. 

2The author have sequened the composition of microbiota of bile and colon feces, the results is interesting, however, it have some different in the 3 kinds of parasite, so how we understanding ,it is better to described in the discussion. 

3 whether the author use hamaster as the animal model need to be described in the manuscript.

PLOS authors have the option to publish the peer review history of their article (what does this mean?). If published, this will include your full peer review and any attached files.

Reviewer #1: No

Reviewer #2: No

Reviewer #4: Yes: OK
---

## [Decision Letter · Decision Letter 1]

20 Jan 2023

Dear Dr. Mariya Y Pakharukova,

We are pleased to inform you that your manuscript 'Opisthorchis viverrini, Clonorchis sinensis and Opisthorchis felineus liver flukes affect mammalian host microbiome in a species-specific manner' has been provisionally accepted for publication in PLOS Neglected Tropical Diseases.

Before your manuscript can be formally accepted you will need to complete some formatting changes, which you will receive in a follow up email. A member of our team will be in touch with a set of requests and in **that occasion please insert the final amendment requested by Reviewer #1**

Best regards,

Wannaporn Ittiprasert, Ph.D

Academic Editor

Francesca Tamarozzi

Section Editor

The manuscript ID PNTD-D-22-01019R1 "Opisthorchis viverrini, Clonorchis sinensis and Opisthorchis felineus liver flukes affect mammalian host microbiome in a species-specific manner" reports the novel knowledge of microbiome in liver flukes as O. viverrini and C. sinensis. The experimental design and statistic analysis are well plan to suport the hypothesis investigation. The revised version reaches the satisfy to all comments, suggestions of  reviewers especially method section. The basic knowledge from this field study would be useful to the field for further and/or advanced studies to understand the microbiome in macroparasite, microparasite traveling from host to another host etc. The current version satisfies the criteria established by PloS NTD is beneficial for researchers in the field as well as those in related fields.

Reviewer's Responses to Questions

**Key Review Criteria Required for Acceptance?**

**Methods**

-Are the objectives of the study clearly articulated with a clear testable hypothesis stated?

-Is the study design appropriate to address the stated objectives?

-Is the population clearly described and appropriate for the hypothesis being tested?

-Is the sample size sufficient to ensure adequate power to address the hypothesis being tested?

-Were correct statistical analysis used to support conclusions?

-Are there concerns about ethical or regulatory requirements being met?

Reviewer #1: Yes, the authors clearly response and edit manuscript according to the suggestion.

Reviewer #2: All comments have been addressed.

Reviewer #4: Yes

The study design appropriate to address the stated objectives.

The population clearly described and appropriate for the hypothesis being tested.

The sample size sufficient to ensure adequate power to address the hypothesis being tested?

The statistical analysis used to support conclusions is correct

The concerns about ethical or regulatory requirements being met.

**Results**

-Does the analysis presented match the analysis plan?

-Are the results clearly and completely presented?

-Are the figures (Tables, Images) of sufficient quality for clarity?

Reviewer #1: Yes. But please add the discussion about H. pylori, as in the response to the reviewer, into discussion.

Reviewer #2: All comments have been addressed.

Reviewer #4: Yes

**Conclusions**

-Are the conclusions supported by the data presented?

-Are the limitations of analysis clearly described?

-Do the authors discuss how these data can be helpful to advance our understanding of the topic under study?

-Is public health relevance addressed?

Reviewer #1: Yes

Reviewer #2: The authors have satisfactorily addressed all of my review comments. Thank you for your comprehensive response.

Reviewer #4: Yes

**Editorial and Data Presentation Modifications?**

Reviewer #1: Accept with minor revision (as above)

Reviewer #2: The authors have adequately addressed to comments mentioned in a previous round of review.

This manuscript is now acceptable for publication.

Reviewer #4: Accept

**Summary and General Comments**

Reviewer #1: The authors are well response to the suggestion.

Reviewer #2: (No Response)

Reviewer #4: The author have answered my question well, i think it can be accept.

PLOS authors have the option to publish the peer review history of their article (what does this mean?). If published, this will include your full peer review and any attached files.

Reviewer #1: No

Reviewer #2: No

Reviewer #4: **Yes: **Sun Xi

---

## [Editor Report · Acceptance letter]

9 Feb 2023

Dear Dr. Pakharukova,

We are delighted to inform you that your manuscript, "*Opisthorchis viverrini, Clonorchis sinensis and Opisthorchis felineus* liver flukes affect mammalian host microbiome in a species-specific manner," has been formally accepted for publication in PLOS Neglected Tropical Diseases.

Best regards,

Shaden Kamhawi

co-Editor-in-Chief

Paul Brindley

co-Editor-in-Chief
